# Paper-Based ZnS:Cu Alternating Current Electroluminescent Devices for Current Humidity Sensors with High–Linearity and Flexibility

**DOI:** 10.3390/s19214607

**Published:** 2019-10-23

**Authors:** Yaqin He, Mengyao Zhang, Nan Zhang, Danrong Zhu, Chun Huang, Ling Kang, Xiaofeng Zhou, Menghan Hu, Jian Zhang

**Affiliations:** Shanghai Key Laboratory of Multidimensional Information Processing, East China Normal University, 500 Dongchuan Road, Shanghai 200241, China; 51171213022@stu.ecnu.edu.cn (Y.H.); 51181213046@stu.ecnu.edu.cn (M.Z.); 52171213016@stu.ecnu.edu.cn (N.Z.); 51171213080@stu.ecnu.edu.cn (D.Z.); 52151213018@stu.ecnu.edu.cn (C.H.); 52171213015@stu.ecnu.edu.cn (L.K.); xfzhou@ee.ecnu.edu.cn (X.Z.); mhhu@ce.ecnu.edu.cn (M.H.)

**Keywords:** paper-based ACEL, capacitive humidity sensor, ZnS:Cu, flexible, high–linearity

## Abstract

Humidity sensors are indispensable for various electronic systems and instrumentations. To develop a new humidity sensing mechanism is the key for the next generation of sensor technology. In this work, a novel flexible paper-based current humidity sensor is proposed. The developed alternating current electroluminescent devices (ACEL) consist of the electroless plating Ni on filter paper and silver nanowires (AgNWs) as the bottom and upper electrodes, and ZnS:Cu as the phosphor layer, respectively. The proposed humidity sensor is based on ACEL with the paper substrate and the ZnS:Cu phosphor layer as the humidity sensing element. The moisture effect on the optical properties of ACELs has been studied firstly. Then, the processing parameters of the paper-based ACELs such as electroless plated bottom electrode and spin-coated phosphor layer as a function of the humidity-sensitive characteristics are investigated. The sensing mechanism of the proposed sensor has been elucidated based on the Q~V analysis. The sensor exhibits an excellent linearity ($R^{2}=0.99965$) within the humidity range from 20% to 90% relative humidity (RH) and shows excellent flexibility. We also demonstrate its potential application in postharvest preservation where the EL light is used for preservation and the humidity can be monitored simultaneously through the current.

## 1. Introduction

Humidity monitoring is widely applied in the paper industry, tobacco industry, microelectronics, intelligent home, etc. Therefore, the development of fast and effective humidity detection technology has always been a concern for the community. So far various types of humidity sensors based on capacitance [1,2,3], resistance [4,5,6], current [7], quartz crystal microbalance [8,9,10], and optical detection [11,12,13] have been developed. The continuously growing demand for humidity monitoring requires the sensing devices with not only good electrical properties but also high mechanical robust capabilities.

Paper has been widely used in flexible electronic devices due to its attractive characteristics, such as chemically and mechanically stable under atmospheric conditions, low-cost, recyclable, light-weighted, and environmentally benign. For example, filter paper itself is a good sensing element candidate for the detection of gases and humidity due to its high flexibility and inherent porous structure, rough surface which facilitates to absorb the vapors. It was reported that it provides the large surface area comparable to the nonporous plastics [14]. In a humidity capacitive sensor, water vapor diffused, and absorbed by the dielectric material lead to the variation of dielectric constant. Since the “dry” dielectric constant is significantly lower than the “wet” one, the electrical capacitance increases as the dielectric absorbs water, hence reflecting the humidity of the surrounding environment. In addition, in the field of paper-based electronics paper was also used as a substrate or active element for sensor development [15,16,17]. For example, various semiconducting compounds have been deposited on filter paper for the detection of gases and humidity [18,19,20,21]. ZnS is one of the most crucial II to VI semiconductor compounds [22] because of its unique photoelectronic properties. Quantities of studies showed that ZnS-based materials have high sensitivity to moisture [23,24,25]. Therefore, it provides the possibility to construct the high-precision humidity sensor.

ZnS-based materials are also potential for luminescence. ACELs are very promising due to their intrinsic ability of uniform light emission, flexible architecture, low heat generation, and low power consumption [26,27,28,29]. Paper-based electroluminescent device has been proposed [30] and in most cases the dielectric layer of ACELs is made of barium titanate or some other materials. It was reported that the luminescence of ZnS is influenced by the moisture [31]. Inspired by this concept, in this context, based on the detailed study of the effect of humidity on the electrical and optical properties of ACELs, we propose a novel flexible current-type humidity sensor. The ACELs with filter paper as the dielectric and ZnS:Cu as the active layer were fabricated. The humidity was measured through the current change due to the water vapor adsorption by both the filter paper and ZnS:Cu active layer. The humidity characteristics and the sensing mechanism were studied and discussed. The humidity sensor based on ACELs provides a new route for humidity detection.

## 2. Experimental

### 2.1. Materials and Reagents

AgNWs solution (5 mg/mL) was purchased from Aladdin Industrial Corporation (Shanghai, China). ZnS/Cu phosphor microparticles with radius of 5~7 μm and PB glue were synthesized by Keyan Phosphor Technology Company (Shanghai, China). The amount of Cu activator in the phosphor is around 0.1 wt% ZnS, and the size of ZnS:Cu powder is around 4–10 μm. Filter paper was purchased from Whatman.

### 2.2. Fabrication of Paper-Based ACELs

The innovative ACELs based on filter paper substrates were fabricated via the all-solution processes, as shown in Figure 1a. First, nickel thin film electrode on one side of filter paper was achieved through electroless plating nickel [32]. Next, with Ni-coated filter paper as substrate, the ACELs were prepared by spin-coating processes consequently: To obtain the phosphor layer, the mixture of ZnS:Cu powder, and PB glue with a weight ratio of 2:1 was spin-coated onto the electrode-coated filter paper, and baked at 120 °C for 30 min. The spin-coating speed varied from 1000 to 4000 rpm to achieve the different thicknesses. At last, the AgNWs electrode was spin-coated onto the active phosphor layer with 1500 rpm, following baking at 120 °C for 5 min. Before each spin-coating process, plasma treatments have been employed to enhance the hydrophilic properties of surface. Additionally, the flexible ACEL with a structure of AgNWs/phosphor layer/filter paper/Ni was obtained. The final ACELs structure is schematically shown in Figure 1b.

### 2.3. Characterization and Measurement

The structural and morphological properties of as-received ACELs were examined by scanning electron microscopy (SEM, Hitachi S4800, Tokyo, Japan). The power source for ACELs was a homemade Alternating Current (AC) system which could provide 1000 V with the frequency of ~10 kHz. The luminance of ACELs was measured using a calibrated broad-band optical luminance meter (GREEN-Wave-VIS StellarNet, Tampa, FL, USA). For relative humidity testing, the RH-controlled environment was realized in a custom-made system consisting of an adaptable closed humidity chamber, a high precise spectrum, and a standard hygrometer, as shown in Figure 1c [33,34]. The ACEL was fixed inside the humidity chamber which had the inlet and outlet valves for the gas flow control. The pure dry nitrogen was used as the carrier gas. The input gas was a mixture of dry and wet nitrogen. The wet gas was generated by bubbling pure dry nitrogen in saturated aqueous solution of K_2_SO_4_ and in principle the maximum RH value yielded can be ~97% at room temperature at 25 °C. By adjusting the flow ratio between the dry/wet gases, a wide humidity range, ~20–90%, could be obtained. The current of ACELs was measured by a digital multimeter (17B+ FLUKE, Shanghai, China). During the humidity experiment, we measured the changes of current and light intensity with the % RH simultaneously. To precisely calibrate the sensing performance, we used a commercial humidity sensor (LX8013, LeXiang, Guangzhou, China, 0–99%) to monitor the chamber. During testing, each humidity input was kept constant for 10 min to obtain steady humidity environment. After test was finished, the flow rate of wet nitrogen was turned back to zero, and then the pure nitrogen was used to flush the chamber until the low humidity environment reached.

## 3. Results and Discussion

### 3.1. Characterizations of Paper-Based ACELs

Figure 2 gives the SEM photos of a cross-sectional AgNWs/ZnS:Cu@PB glue/filter paper/Ni structure and the corresponding element mappings. From Figure 2a, it can be clearly seen that the Ni film is successfully grown on one side of the filter paper. The XRD spectrum of prepared Ni is shown in Appendix A. In addition, the phosphor layer is located on the other side. Ni film electrode separates from the phosphor layer through the pristine filer paper. The SEM images of Ni and Zn:Cu phosphor are shown in Appendix A. Therefore, except for the substrate, the filter paper can act as the dielectric layer for ACELs simultaneously. The top electrode obtained from AgNWs is shown in the magnified picture in Figure 2b. Elemental distribution maps obtained by SEM are shown in Figure 2c. The distribution of a particular element including Ag, Zn, S, C, O, and Ni corresponds to the structure of the prepared EL device. Moreover, the EDS mapping was also carried out to identify the composition of the prepared paper-based ACEL shown in Appendix A. The thickness of the Ni film and phosphor layer can be adjusted by the spin-coating speed and electroless Nickel (EN) time, respectively.

### 3.2. Luminescent Properties of Paper-Based ACELs

Alternating current-driven inorganic powder electroluminescent devices were fabricated on the filter paper substrate via the all-solution processes. Nickel bottom electrodes were prepared from electroless plating solution, and the simple spin-coating method was used to fabricate the active layer and the top electrode. So, the whole fabrication processes were simplified greatly. In order to optimize the EL performance, we controlled the mixing ratio of the emitting-dielectric composite, which was composed of ZnS:Cu powder and an organic binder-PB glue. The EL sample with a phosphor-to-binder mixing ratio of 2:1 exhibited the highest EL performance. Here, it is worth noting that filter paper has multifunctional including: First, it acts as the carrier substrate for nickel electrode. Second, it uses the dielectric layer for ACELs. At last, the porous paper is a good absorber. Therefore, the luminance of paper-based ACELs can be used as the indicator to sense the humidity since the environmental water vapor can influence the electrical properties like dielectric constant remarkably.

The characteristics of the ACELs were measured by applying an AC voltage under ambient conditions. It is well known that the ZnS:Cu phosphor can give blue-green emissions corresponding to the transitions between Cu and ZnS materials. Figure 3a shows the various patterned EL images of ACEL fabricated on polyimide and paper substrate. The novel ACEL fabricated on filter paper exhibits a blue-green light emission with the EL excitation wavelength of ~506 nm shown in Figure 3c. Figure 3c also displays the luminescent spectra of EL device operated under different humidity levels. It demonstrates that the central emission wavelength is not affected by the humidity. However, the luminescent intensity of the device increases with RH at the fixed voltage and frequency. Under different humidity levels, the CIE (Commission International edel Eclairage 1931) diagram is used to indicate the variations in the fluorescence color from the initial green (0.304, 0.320) to green (0.282, 0.324) shown as Figure 3b. Figure 3d shows the time dependent properties of the AC Root Mean Square (RMS) current under different humidity RH levels. All these results indicate that both the luminescent and the electrical properties of ACELs are affected by the humidity and are possible to be used as the humidity indicator. However, both the luminescent intensity and the CIE color coordinate are very difficult to be calibrated and used to construct the humidity sensor. In terms of the electrical properties, in fact, Young et al. [31] had studied the electrical dependence of ZnS thin films exposed to H_2_O. Therefore, it is unquestionable to construct the humidity sensor via the electrical response of ZnS:Cu. More importantly, there also exists the potential to develop the new humidity detection technology which can integrate and employ both the electrical and optical responses simultaneously.

### 3.3. Current Humidity Sensors from Paper-Based ACELs

In principle, ACEL can be regarded as the serial of two capacitors. First, the capacitance response of ACELs without AC signal source applied was studied by a precision LCR meter at varied relative humidity values ranging from 20% to 90%. Figure 4a demonstrates the response of EL device towards relative humidity at room temperature at 25 °C for three tests. It can be seen that the nonlinear response is obvious. When the RH value is below 60%, the capacitance of the ACEL slightly increases with the RH. An obvious increase of the capacitance with RH is found when the humidity level is bigger than 60% RH. The capacitance variations of ACELs can be associated with the adsorption and desorption of water molecules on both the dielectric layer and the phosphor layer. The sensing mechanism of ACELs response behaviors for humidity sensing will be discussed in the next section. Obviously, in terms of new-type humidity sensor research and development, this nonlinearity response is a disaster.

In order to improve the linearity, the characteristics of ACELs excited by the AC signal source in different humidity levels was studied. In order to simplify the testing, we measured the effective values of AC current instead of the capacitance. The digital multimeter in serial with the ACEL was employed as shown in Figure 1c which could record the real-time AC current values through LabVIEW ware. For our ACELs structure, the factors that influence the luminescent properties mainly include the dielectric layer thickness, the nickel electrode, and the active layer thickness. Among them, the first two are closely related.

First, the effect of nickel metallization on humidity-sensing characteristics was investigated. Figure 4b shows the relationship between the current and the EN time since the EN time can influence the bottom Ni electrode greatly. Here, the spin-coating speed of phosphor layer and the top electrode layer were fixed. Different EN time was selected in order to change the conductivity of the Ni electrode and the thickness of dielectric layer simultaneously. Apparently, the longer the EN time, the smaller the sheet resistance is. In all cases, the current increases with the humidity level accordingly. In addition, it can be seen that the correlation curve of I~RH shifts to higher current when the plating time increases. The result also indicates that plating time shows non-significant effect on the linearity of I~RH curves.

The current responses of ACELs with different ZnS:Cu phosphor layers were studied. The EN time was fixed at 10 min. The thickness was controlled by adjusting the spin-coating speed. The relationship between the current of ACELs with different phosphor layers and the humidity levels is shown in Figure 4c. It can be seen that all curves tend to increase with the humidity level monotonically. By changing the thickness of phosphor layer, the linearity of I~RH is improved remarkably. Additionally, the device fabricated with 1000 and 2000 rpm shows a better linearity than those of 3000 and 4000 rpm. Especially, when the rate is 2000 rpm, the device exhibits the perfect linear I~RH curve.

Through linear fitting the curve obtained above, the linear regression equation can be expressed as y=6.50x+1.17 with correlation factor $R^{2}=0.99965$. If we define the sensitivity as the slope of the I~RH curve, the values of sensitivity are 6.50 μA/RH. In order to examine the repeatability, three repeated tests were conducted under the same condition and the result is shown in Figure 4d. It is found that the device has the superior repeatability and the maximum repeatability error reached ±6.25% and 85% RH. The results obtained above indicate that the device can be used as a probe to monitor the humidity in our living environment and the good sensing repeatability as shown in Figure 4d during the three repeated tests could be an advantage for its potential applications.

The time-dependent current response of the paper-based ACELs was tested by switching the device between the low humidity environment (11.9% RH) and the 80% RH environment. The dynamic current response of the device is shown in Figure 4e. The corresponding response and recovery constants can be extracted. The response time and recovery time (defined as the time for reaching 63% of the steady state) are 3.0 and 2.4 min, respectively. The device exhibits a long response/recovery time due to its stacked structure and the properties of filter paper. There are some strategies to optimize the response/recovery time of the prepared humidity sensor. Firstly, instead of the filter paper, some other paper with thinner thickness and better porosity can be employed. Secondly, the sensor with smaller size such as point shape is beneficial for the fast detection [35]. Additionally at last, the recovery time could be reduced by integrating a heater to the RH sensor. In addition, the reversible response of the sensor was tested and the typical response is given in Figure 4f, showing excellent reversibility of this kind of humidity sensor during cycling 11.9% and 80% RH at room temperature.

### 3.4. Humidity Sensing Mechanism for the ACELs

In principle, the change of AC current is attributed to the change of capacitive reactance in the humidity sensing behavior of paper-based ACELs can be explained roughly based on the electrical properties changes due to the water molecules adsorption/desorption, and adsorption of water vapor is totally responsible for changes in capacitive reactance of the ACELs. The porous and rough surface of the filter paper is favorable to accommodate water vapor which is beneficial to improve the sensitivity of humidity sensor. For a reasonable explanation of the capacitive modulation of the phosphor layer with the RH we have to consider semiconducting properties. It is known that ZnS is thermodynamically unstable in the presence of oxygen and should be converted to ZnO, one can expect oxygen promoted water molecule physisorption. In general, the formation of the SH- groups can be expected for the sulfide surface after interaction with water molecules. However, this effect has not been observed experimentally as a common trend [36]. As a result, the surface pre-adsorbed oxygen plays a main role in formation of an absorbed water layer. To elucidate the working mechanism, the charge/discharge behavior of ACELs was carefully examined. To study the electrical behavior, a common experimental configuration was used, as shown in Figure 5a [37,38]. Moreover, Figure 5b displays the equivalent electric circuit diagram. For every experiment, the voltages $V_{1}\left(t\right)$ and $V_{2}\left(t\right)$ applied on the whole system and the sense capacitor $C_{S}$ were measured and recorded using two high-voltage probes (Tronovo TR9308A, Jiangsu, China and GENTEK G3100, Shanghai, China) and two-channel oscilloscope (Tektronix^®^ TBS 1052B, Shanghai, China), as shown in Figure 5c.

The ACEL is connected in series with a fixed capacitor $C_{s}$. Since the dielectric layer and the phosphor are wide-bandgap insulators with no or few free charges, the ACEL electrically behaves like two capacitors $C_{p}$ and $C_{d}$ in series. $C_{p}$ and $C_{d}$ correspond to the capacitances of phosphor layer and dielectric layer, respectively. When AC is applied, the charge applied to the device is equal to that applied to the sense capacitor $C_{s}$. When the external voltage reaches the threshold voltage, a large number of carriers are injected and the phosphor layer is effectively shorted out. In brief, $C_{p}$ disappears and only $C_{d}$ exists. Therefore, by studying the AC response of the device, we can clearly discern the evolutions of these capacitors. On this basis, we can further study the influence of the humidity on these capacitors. The time-dependent voltage measured across the sense capacitor $C_{s}$ forms the basis for all of the standard electrical measurements. In this study, the AC working frequency and the sense $C_{s}$ were 10 kHz and 0.126 nF, respectively. The external charge-voltage (Q~V) measurement is the most straight-forward standard characterization technique [39,40]. The applied voltage V(t) on ACEL was determined as $V\left(t\right)=V_{1}\left(t\right)-V_{2}\left(t\right)$. Since the charge on the two capacitors connected in series must be equal by Gauß’ Law, the measurable charge $q=V_{2}\left(t\right)\times C_{s}$ on the $C_{s}$ must be equal to the unmeasurable charge on the ACEL. In order to make a comparison clearly, in this paper the charge is further normalized to the device area. A family of Q~V plots were obtained by changing the humidity level and the corresponding results are shown in Figure 5d. From Figure 5d, it can be seen that the areas enclosed by the Q~V plots, i.e., the consumed energy during each cycle testing, gets larger with the increasing humidity level. Moreover, the counter-clockwise progression of Q~V curves is apparent with the increasing of humidity level. The capacitors under different humidity levels were abstracted from the family of Q~V plots and the results are reviewed in Table 1. It can be seen that both $C_{P}$ and $C_{d}$ become larger as the RH increases. In addition, $C_{d}$ increases faster than $C_{P}$. So far we can obtain the humidity sensing mechanism of ACELs: Due to the bigger dielectric constant of water in comparison with the phosphor layer and the filter paper, after the phosphor layer and the dielectric layer uptake the water molecules, $C_{P}$ and $C_{d}$ have increasing tendency. Among them, the latter $C_{d}$ seems to play a greater role. Under the luminescent condition, the capacitor $C_{d}$ doesn’t work. Therefore, the dielectric constant $\varepsilon_{p}$ can be extracted since the electrode area is known and the thickness of phosphor layer can be determined from the SEM in Figure 2a. The extracted $\varepsilon_{p}$ values are also listed in Table 1. It can be seen that with the humidity level increasing, $\varepsilon_{p}$ tends to increase. This is reasonable since the water absorption can increase the dielectric constant of the phosphor layer. The increased dielectric constant lead to the increased $C_{P}$, i.e., the increased charges, thus enhancing the luminance and the AC current as well. This result is also in agreement with the published results. The single crystal ZnS has $\varepsilon_{P}=5.2$ and $\varepsilon_{P}$ raised to 11~12 after exposure to 87% RH [31]. Of course, the variations of $\varepsilon_{p}$ is closely related to lots of factors such as the size of ZnS powders and the glue type, etc. In addition, the analysis also indicates that the consumption power is <2 W in all cases and will slightly increase with the increasing humidity level.

### 3.5. Bending Properties and Application of the Device

The flexibility is essential for humidity sensors applications in electronics. Thus, the static dynamic properties were measured. The results are shown in Figure 6. From Figure 6a, it can be seen that in all cases, the ACELs can still work properly, indicating that the devices have excellent flexible properties. The static bending properties of ACELs such as convexed, normal, and concaved were measured under different humidity conditions. Figure 6b demonstrates that the output I~RH relationship of ACELs exhibits the good linearity under different bending conditions. Under convexed (90°) and concaved (−90°) conditions, the curves deflected from the normal case is probably due to the conductivity variations of AgNWs and Ni electrodes affected by bending. Under the convexed and concaved bending conditions, the fitting equations can be extracted as
y=6.36x+1.38 and y=6.31x+1.38 with the correlation factors $R^{2}=0.99089$, and 0.99651, repectively. The corresponding sensitivity values under different conditions are 6.50 (normal), 6.36 (convexed), and 6.31 μA/RH (concaved), respectively. These results indicate that the performance of the paper-based ACELs is highly stable since the sensitivity keeps almost constant, i.e., the bending shows the non-significant effect on the linearity and the sensitivity. The bending measurement also demonstrates that the EL devices have highly robust and stability and are promising for applications to flexible devices. Figure 6c shows one possible application, where the flexible ACELs can be attached on the inner or outer wall of a glass tube and work at concaved and convexed conditions. In addition, the luminescence of the device motivated by the AC source appears to be somewhat uneven due to the rough structure of the filter paper.

It is understood that low quantities of light can maintain the postharvest quality of vegetables and crops by mitigating senescence, and improving phytochemical and nutrient content in several species [41,42,43]. Additionally, it was reported that the luminescence in a certain wavelength range can not only preserve vegetables but also promote plant growth [44,45,46]. Figure 6d,e displays one possible and potential application of our developed sensors in postharvest preservation. The flexible ACEL can be attached on the container’s wall or wrapped to enclose the vegetable. In this condition, the EL device can offer two functions: To realize the postharvest preservation and to monitor the humidity level simultaneously. It is worth noting that by adjusting the phosphor activator, working voltage and frequency, both the intensity and the wavelength of the flexible ACEL can be modulated.

## 4. Conclusions

We have fabricated the AC-type ACELs with total solution processes on filter paper. It is found that the ACEL sensor exhibits an excellent linearity ($R^{2}=0.99965$) within the humidity range from 20% to 90% RH due to the synergetic function of filter paper and ZnS:Cu. The sensing mechanism for the flexible paper-based ACELs was studied through electrical analysis. Moreover, the bending measurement shows that the paper-based ACELs have excellent flexibility and can work properly under different bending conditions. We also demonstrate its potential application in postharvest preservation for vegetables where the EL light is used for preservation and the humidity level can be monitored simultaneously through the AC current.

## Figures and Tables

**Figure 1 sensors-19-04607-f001:**
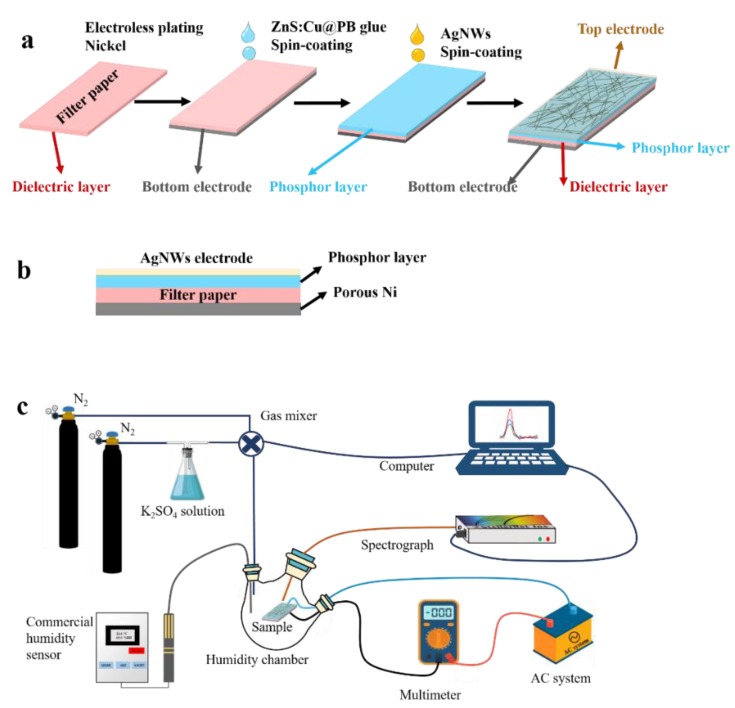
(**a**) Flow chart for paper-based alternating current electroluminescent devices (ACELs) fabrication. (**b**) Schematic image of the device. (**c**) Testing equipment used for optical and electrical characterization of the electroluminescent (EL) devices.

**Figure 2 sensors-19-04607-f002:**
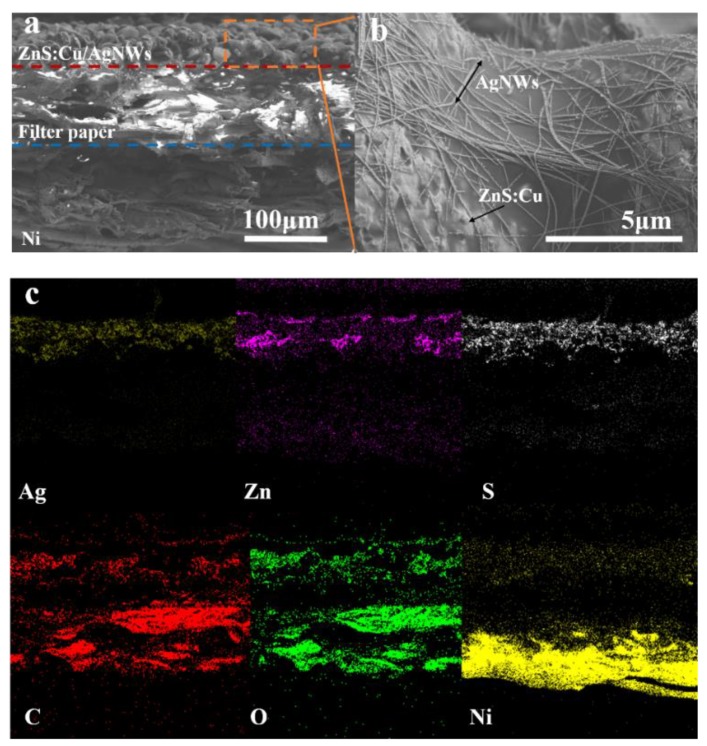
(**a**) Cross-sectional SEM photo of ACEL interface. (**b**) SEM photo of the AgNWs on top. (**c**) Distribution maps of the elements Ag, Zn, S, C, O, and Ni.

**Figure 3 sensors-19-04607-f003:**
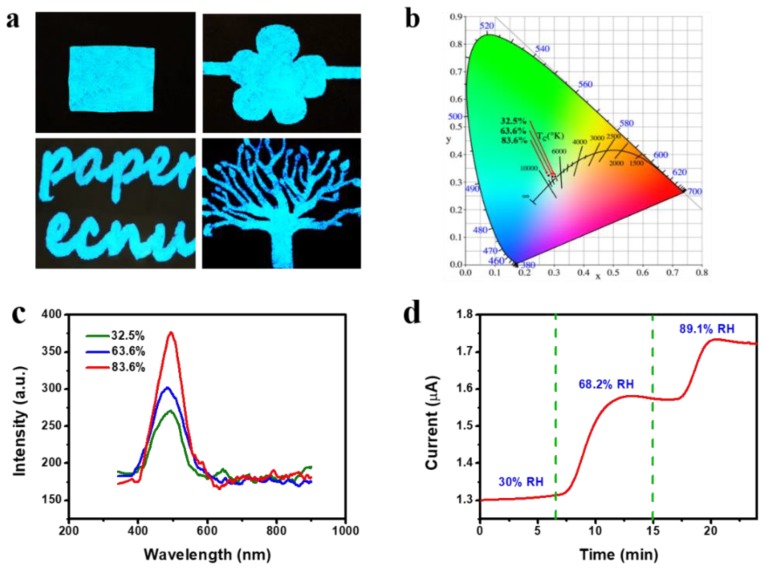
(**a**) EL images of ACEL device fabricated on polyimide. (**b**) Commission International edel’Eclairage (CIE) chromaticity diagram of the (x,y) color ordinates of the device varying with RH. (**c**) EL spectra of the device in different relative humidity (32.5%, 63.6%, 83.6%) at room temperature. (**d**) Current versus time plot at different (%) RH level.

**Figure 4 sensors-19-04607-f004:**
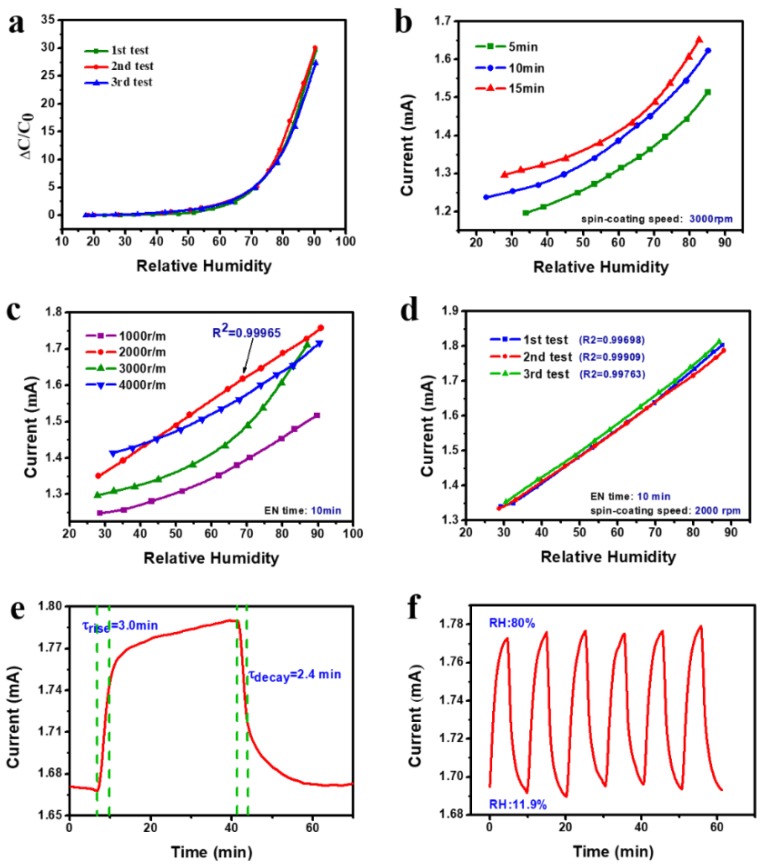
(**a**) Capacitance response to relative humidity in three different measurements with the relative humidity range from 20% to 90%. (**b**) Alternating current behavior of the device with different EN time of bottom electrode-Ni under different humidity levels. (**c**) I~RH curves of paper-based ACELs with different spin-coating speed of phosphor layer. (**d**) Current response to relative humidity in three different measurements. (**e**) Current-time curve of the device from 11.9% to 80% RH estimating response and recovery time. (**f**) Cycling stability of paper-based ACEL from 11.9–80% RH level.

**Figure 5 sensors-19-04607-f005:**
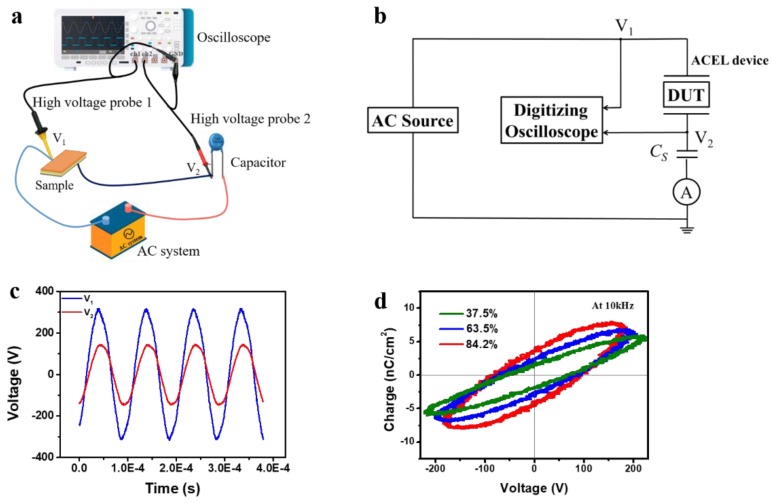
(**a**) Electrical characterization setup and the (**b**) equivalent electric circuit diagram. (**c**) Voltage waveform recorded during the test. (**d**) A family of Q~V curves for paper-based ACELs driven at 10 kHz.

**Figure 6 sensors-19-04607-f006:**
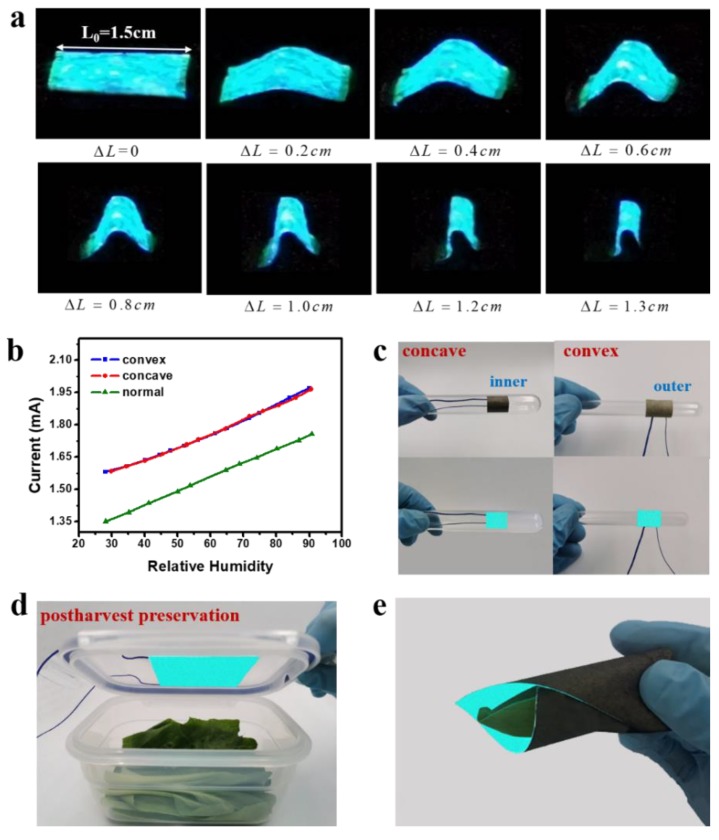
(**a**) Measurements for the bending performance of the ACELs. (**b**) Linear fitting curve of current and humidity from 26% to 90% under different conditions. (**c**) Images of the paper-based ACELs working under different bending conditions. (**d**,**e**) Images of the device working for postharvest preservation.

**Table 1 sensors-19-04607-t001:** Analytic results of the family of Q~V curves.

RH (%)	$C_{d}\;\left(nF\right)$	$C_{P}\;\left(nF\right)$	$C_{total}\;\left(nF\right)$	$\varepsilon_{p}\;\left(F/m\right)$	P (W)
37.5	0.077	0.127	0.048	7.71	1.29
63.5	0.102	0.137	0.059	8.31	1.49
84.2	0.143	0.142	0.071	8.62	1.71

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
