# Peer review of "Paper-Based ZnS:Cu Alternating Current Electroluminescent Devices for Current Humidity Sensors with High–Linearity and Flexibility"

_sensors, 2019, doi:10.3390/s19214607_

Round 1

Reviewer 1 Report

It does not match well with the fabrication process and 2.2 Fabrication of Paper based ACELs in Figure 1 (a). Therefore the picture needs complementary. (In this article, spin coating is done, but in the picture, it is expressed as spraying water droplets.) If your fabrication process says it is liquid, then all surfaces must remain hydrophilic for that liquid. Could you tell me in detail in hydrophilic surface?  In 208 line “The good sensing repeatability as shown by the highly repeatable sensing curves during 208 cycling tests could be an advantage for its potential applications” Please express how many repeatable numbers.

Author Response

comments:

It does not match well with the fabrication process and 2.2 Fabrication of Paper based ACELs in Figure 1 (a). Therefore the picture needs complementary. (In this article, spin coating is done, but in the picture, it is expressed as spraying water droplets.) If your fabrication process says it is liquid, then all surfaces must remain hydrophilic for that liquid. Could you tell me in detail in hydrophilic surface?  In 208 line “The good sensing repeatability as shown by the highly repeatable sensing curves during 208 cycling tests could be an advantage for its potential applications” Please express how many repeatable numbers.

Reply:  

Thanks for you good advice. The Figure 1(a) had been modified following the reviewer’s comment.

       The additional O2 plasma treatments have been employed to enhance the hydrophilic properties of surface. The corresponding content had been added in the manuscript as below “Before each spin coating process, plasma treatments have been employed to achieve hydrophilic surface.” (in Page 2, Line 77-78)

       Three repeated tests have been done and the results have been shown in Figure 4(d).  And Figure 4(f) shows the excellent reversibility of this kind of humidity sensor during cycling between 11.9% and 80% RH at room temperature. The authors had re-written this section. And the corresponding description “The results obtained above indicate that the device can be used as a probe to monitor the humidity in our living environment and the good sensing repeatability as shown in Figure 4(d) during the three repeated tests could be an advantage for its potential applications.” (in Page 6, line 204-205) and “In addition, the reversible response of the sensor was tested and typical response is given in Figure 4(f), showing excellent reversibility of this kind of humidity sensor during cycling 11.9% and 80% RH at room temperature.” (in Page #6, line 215-218 ) was added and marked in revised manuscript.

Reviewer 2 Report

In the manuscript (sensors-616436) entitled as " Paper-based ZnS:Cu Alternative Current Electroluminescent Devices for High–linearity and Flexibility Current Humidity Sensors", a novel flexible paper-based current humidity sensor is proposed. The sensor exhibits an excellent linearity within the humidity range from 20 to 90% RH and shows excellent flexibility. In general, the idea is interesting but I can recommend this paper to be published in Sensors after minor revision.

Comments

As shown in Figure 4(e), the response time and recovery time are 3.0 and 2.4 min, respectively. And the authors claimed that the stacked structure and the properties of filter paper resulted in the long response/recovery time. Can the author provide a clue to reduce the response time of the proposed sensor. I suggest the authors mention the MNF based RH sensor with a response time of 70 ms (Optics Express 16, 13349-13353.).

2 The author claimed that the sensor exhibited an excellent linearity within the humidity range from 20 to 90% RH and shows excellent. Can the sensor work in a high RH condition? Recently, a Nafion film based RH sensor that can work in a RH range of 30-100% (IEEE Sensors Journal 19, 9229-9234.). I suggest the authors mention this paper and review the following works, which shows other interesting techniques for flexible RH sensing.

Porous Ionic Membrane Based Flexible Humidity Sensor and its Multifunctional Applications. Adv. Sci. 2017, 1600404, 4(5):1-7. 

3 Humidity sensing mechanism of the ACELs should be explained more clearly. Also, I suggest the author compare the performance (e.g., response time, working range, sensitivity, resolution) the proposed RH sensor with the typical reported RH sensors in a table.

Author Response

Comments:

In the manuscript (sensors-616436) entitled as " Paper-based ZnS:Cu Alternative Current Electroluminescent Devices for High–linearity and Flexibility Current Humidity Sensors", a novel flexible paper-based current humidity sensor is proposed. The sensor exhibits an excellent linearity within the humidity range from 20 to 90% RH and shows excellent flexibility. In general, the idea is interesting but I can recommend this paper to be published in Sensors after minor revision.

1) As shown in Figure 4(e), the response time and recovery time are 3.0 and 2.4 min, respectively. And the authors claimed that the stacked structure and the properties of filter paper resulted in the long response/recovery time. Can the author provide a clue to reduce the response time of the proposed sensor. I suggest the authors mention the MNF based RH sensor with a response time of 70 ms (Optics Express 16, 13349-13353.).

Reply: Thanks for your good advice. It is an important and serious question. There are some methods could be used to improve the response/recovery time. The corresponding description as below had been added “There are some strategies to optimize the response/recovery time of the prepared humidity sensor. Firstly, instead of the filter paper, some other paper with thinner thickness and better porosity can be employed. Secondly, the sensor with smaller size such as point shape is beneficial for the fast detection. And at last, the recovery time could be reduced by integrating a heater to the RH sensor.” (in Page #6, Line 211-215).

2) The author claimed that the sensor exhibited an excellent linearity within the humidity range from 20 to 90% RH and shows excellent. Can the sensor work in a high RH condition? Recently, a Nafion film based RH sensor that can work in a RH range of 30-100% (IEEE Sensors Journal 19, 9229-9234.). I suggest the authors mention this paper and review the following works, which shows other interesting techniques for flexible RH sensing. Porous Ionic Membrane Based Flexible Humidity Sensor and its Multifunctional Applications. Adv. Sci. 2017, 1600404, 4(5):1-7.

Reply: That’s a good question. In fact, the original idea to develop this kind of sensor is for low or moderate humidity level detection. And the high humidity level is harmful for the sensor. For example, the luminescent intensity of sensor will be lowered. following the reviewer’s advice, the corresponding references had been added.

3) Humidity sensing mechanism of the ACELs should be explained more clearly. Also, I suggest the author compare the performance (e.g., response time, working range, sensitivity, resolution) the proposed RH sensor with the typical reported RH sensors in a table.

Reply: Thanks for your good advice. The humidity sensing mechanism of prepared ACELs is important and also complicated. To explained more clearly, we discussed the sensing mechanism of ZnS in detail. “The humidity sensing behavior of paper-based ACELs can be explained roughly based on the electrical properties changes due to the water molecules adsorption/desorption, and adsorption of water vapor is totally responsible for changes in capacitive reactance of the ACELs. The porous and rough surface of the filter paper is favorable to accommodate water vapor which is beneficial to improve the sensitivity of humidity sensor. For a reasonable explanation of the capacitive modulation of the phosphor layer with the RH we have to consider semiconducting properties. It is known that ZnS is thermodynamically unstable in the presence of oxygen and should be converted to ZnO, one can expect oxygen promoted water molecule physisorption. In general, the formation of the SH- groups can be expected for the sulfide surface after interaction with water molecules. However, this effect has not been observed experimentally as a common trend. As a result, the surface pre-adsorbed oxygen plays a main role in formation of an absorbed water layer. “And the corresponding description was added into revised manuscripts (in Page #8, line 228-239).

In fact, the sensor developed in this paper is a new type of sensor. It is a bimodal sensor which can work under two mechanisms. So the similar sensors are seldom found. And it is difficult to make a comparison among the existing humidity sensors.

Ref.

Year

Authors

Sensing material

Range

(%RH)

Sensitivity

Response time

Bimodal

Flexibility

Linear relation(R2)

[1]

2012

Zhang et al.

No coating (PMMA

polymer cladding)

30-90

33.6 pm/%RH

7min

No

No

-

[2]

2013

Xia et al.

Hydrogel

40-90

0.196 dB/%RH

10s-1min

No

No

0.9932

[3]

2010

Ding et al.

PI

30-80

~2 pm/%RH

-

No

No

-

[4]

2010

Wang et al.

Ta2O5

0-100

3.3 pf/R(0-75)

83.2 pf/RH(80-100)

-

No

No

0.9210/0.9980

[5]

2010

Kim et al.

PI

10-90

506 fF/%

RH

6s

No

No

-

[6]

2017

Li et al.

PIM

6.42-93.54

-

0.4s

No

Yes

0.9067

[7]

2019

Cai et al.

Crystal violet doped Nafion film

30-100

1.4%/%RH

150ms

Yes

No

-

This paper

2019

He et al.

Paper and ZnS

20-90

6.5 uA/%RH

3min

Yes

Yes

0.9997

Reviewer 3 Report

Attached

Author Response

comments:

He et al. demonstrated the flexible paper-based current humidity sensor. This sensor is based on ACEL with paper substrate and ZnS:Cu phosphor layer as the humidity sensing element. Interestingly, sensor shows an excellent linearity (R2 = 0.99965) in the humidity range from 20 to 90% RH and very good flexibility. Authors also demonstrated the potential application of this sensor in postharvest preservation where the electroluminescence is used for preservation. In my opinion, this is a very interesting work and deserves publication. I recommend the

publication of this article in Sensors journal of international repute; however authors need to address the following comment:

(i) In Figure 4 (c), the current versus RH graph for spin-coating speed of 3000 r/m is convex in nature as compared to other spin-coating speeds. Authors should explain this.

Reply: Thanks for your positive comments on our research work. The curve for spin-coating speed of 3000r/m is convex as compared to other spin-coating speeds can be explained as the relative capacitive reactance of the phosphor layer and filter paper was changed remarkably by the spin-coating.